# RUB: Evaluating Residual Knowledge in Unlearned Models

## Abstract

Machine Unlearning (MUL) has emerged as a key mechanism for privacy protection and content regulation, yet current techniques often fail to guarantee the complete removal of sensitive information. While most existing works focus on verifying the execution of unlearning, they overlook the critical question of whether models remain robust against adversarial attempts to recover forgotten knowledge. In this work, we advocate for the principle of Robust Unlearning, which requires models to be both indistinguishable from retrained counterparts and resilient against diverse adversarial threats. To instantiate this principle, we propose a unified benchmark, RUB (Robust Unlearning Benchmark), that systematically evaluates the robustness of unlearning algorithms across classification, image-to-image reconstruction, and text-to-image synthesis. Within this framework, we introduce the Unlearning Mapping Attack (UMA) as a generalizable method to detect residual information, and demonstrate how existing attack strategies can be adapted into this framework as long as they conform to the generic UMA framework. Our experiments across discriminative and generative tasks reveal that state-of-the-art unlearning methods remain vulnerable under these evaluations, even when passing standard verification metrics. By positioning robustness as the central criterion and providing a benchmark for adversarial evaluation, we hope RUB paves the way toward more reliable and secure unlearning practices. The codebase and model checkpoints in RUB will be published.

## 1 Introduction

As deep learning models grow increasingly data-dependent, concerns over privacy and data security have intensified. In response to privacy regulations such as the General Data Protection Regulation (GDPR) (Regulation, 2018) and the California Consumer Privacy Act (CCPA) (Pardau, 2018), Machine Unlearning (MUL) has emerged as a potential solution to selectively remove specific data from trained models, allowing for the "right to be forgotten." Beyond privacy concerns, content regulation has become another key motivation for machine unlearning (Kurmanji et al., 2024; Shumailov et al., 2024). To remove impermissible knowledge, such as unlicensed copyrighted material (Yao et al., 2023), malicious information (Yao et al., 2023), or harmful capabilities (Shumailov et al., 2024) from models, machine unlearning ensures that the unlearned models align with ethical and legal standards. To date, MUL techniques have demonstrated strong performance in eliminating the influence of specific data on both privacy-sensitive and content-sensitive tasks (Warnecke et al., 2021; Li et al., 2024; Graves et al., 2021; Tarun et al., 2023; Golatkar et al., 2020; Liu et al., 2022).

Since MUL handles sensitive data, it is inherently vulnerable to adversarial attacks. Prior research has shown that modifying the unlearning process or manipulating training data, such as injecting noise or backdoors, can undermine its effectiveness Thudi et al. (2022); Qian et al. (2023); Liu et al. (2024); Zhang et al. (2024a). However, beyond these conventional attacks, a new category of post-unlearning adversarial attacks has emerged, targeting residual information left in unlearned models (Zhang et al., 2025; Tsai et al., 2024; Han et al., 2024; Pham et al., 2023; Yuan et al., 2024). Though these attacks are particularly designed for diffusion models (DMs) or large language models (LLMs) to fail content erasure, they expose a fundamental vulnerability: unlearned models often retain traces of the forgotten data, which adversarial probes can exploit to resurface unlearned information. In other words, attackers can craft adversarial inputs that recover forgotten knowledge, effectively negating the unlearning process.

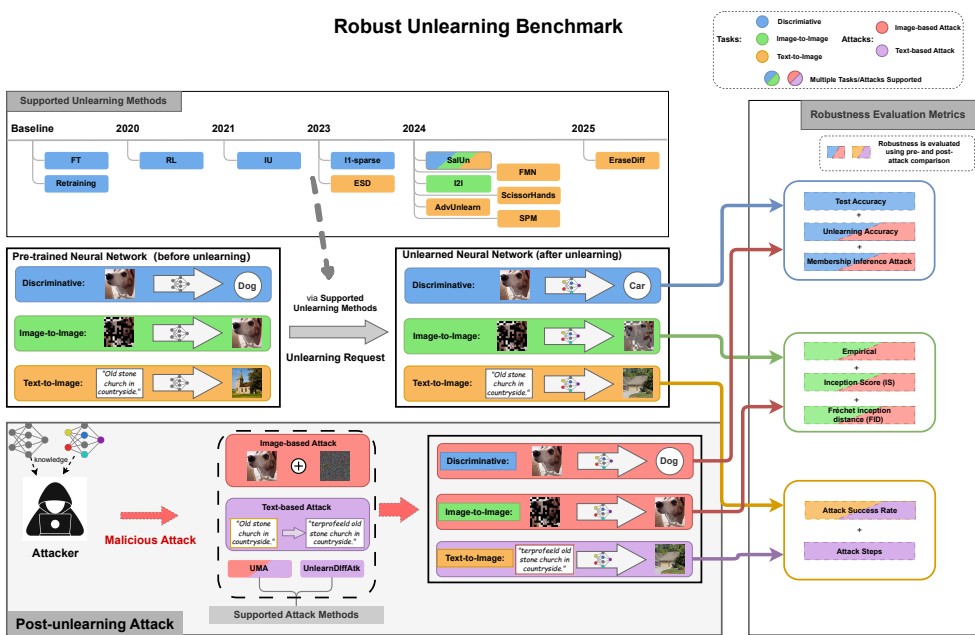

Figure 1: Diagram of our robust unlearning benchmark. RUB covers three common unlearning tasks: classification, image-to-image reconstruction, and text-to-image synthesis, each reflecting distinct unlearning objectives. It supports multiple state-of-the-art unlearning algorithms, task-specific white-box adversarial attack implementations, and appropriate evaluation metrics. Our benchmark provides a unified and extensible framework to assess the robustness of unlearning algorithms against malicious recovery of unlearned information.

To evaluate MUL effectiveness and robustness, verification methods typically fall into attack-based empirical evaluation and process-based reproducibility checks. The former tests an unlearned model's resistance to adversarial threats by either extracting forgotten information (Fredrikson et al., 2015; Shokri et al., 2017; Carlini et al., 2022) or injecting backdoors to deceive the model (Sommer et al., 2020; Gao et al., 2024; Guo et al., 2024). The latter, inspired by Proof of Learning (PoL) (Jia et al., 2021), logs the unlearning process, allowing users or auditors to verify whether unlearning was executed (Zhang et al., 2024a). Despite their usefulness, these methods focus primarily on whether unlearning was performed, rather than ensuring that all traces of the unlearned information have been irreversibly removed.

To address this critical gap in MUL, we introduce the first framework, namely Robust Unlearning Benchmark (RUB), that systematically evaluates the robustness of machine unlearning against post-unlearning adversarial information resurface over various computer vision tasks. As demonstrated in Fig. 1, RUB covers 15 unlearning algorithms, 3 computer vision tasks (i.e. image classification, image-to-image reconstruction, and text-to-image synthesis) over diverse benchmarking datasets, and different post-unlearning adversarial probing methods. At its core, RUB defines a unified protocol for evaluating whether forgotten information can be recovered under attacks. Particularly, we propose a generic Unlearning Mapping Attack (UMA) as a modular adversarial probing tool, and instantiate it across three major task settings. Our benchmark includes multiple evaluation metrics tailored to different tasks and reveals striking differences in the robustness of existing unlearning methods. The benchmarking results on the 15 unlearning methods and our in-depth discussions highlight potential research directions in the future. Our contributions are summarized as follows:

- We introduce the first benchmark to systematically evaluate unlearning algorithms under post-unlearning adversarial recovery attacks across diverse computer vision tasks.

- We have created a unified evaluation protocol and well-structured code-base that supports 15 state-of-the-art unlearning algorithms on post-unlearning adversarial robustness.

- We design UMA, a general-purpose adversarial attack that can be instantiated across tasks, providing an empirical verification tool to assess whether unlearning methods completely eliminate residual knowledge.
- We have conducted a thorough analysis of various unlearning algorithms in our benchmark, which will inspire researchers to develop more robust unlearning algorithms.

## 2 RELATED WORK

**Machine Unlearning.** Machine Unlearning (Cao & Yang, 2015) was introduced to remove specific data influences from models for privacy and security. The most straightforward approach, retraining from scratch, has become impractical for large models like LLMs due to their high computational cost. To address this, exact and approximate unlearning are proposed. Exact unlearning (Golatkar et al., 2020; Bourtoule et al., 2021) aims to make an unlearned model indistinguishable from one retrained from scratch without the forgotten data. Given scalability challenges, approximate unlearning offers a more practical alternative by relaxing constraints to improve efficiency while maintaining acceptable performance. First-order methods use Taylor series expansions to update model parameters, while second-order methods incorporate the inverse Hessian matrix for more precise adjustments (Warnecke et al., 2021). Recently, SalUn (Fan et al., 2024) improved stability and accuracy by using a weight saliency map to update parameters at varying rates. Despite these advancements, existing methods remain vulnerable to sophisticated attacks that extract forgotten information, raising critical privacy and security concerns.

**Attacks to MUL.** Machine unlearning is inherently vulnerable to adversarial attacks, as evidenced by prior research on malicious attempts against MUL. For instance, adversarial text prompts and Concept Inversion attacks are particularly investigated to undermine DMs for content erasure (Zhang et al., 2025; Tsai et al., 2024; Han et al., 2024; Pham et al., 2023). Targeting more generic unlearning scenarios, Qian et al. (2023) injects targeted noise into forget samples, leading the unlearned model to fail in classification tasks. Liu et al. (2024) induces backdoor behavior in a model through the standard MUL process with selected data. Thudi et al. (2022); Zhang et al. (2024a) leverage techniques from Data Ordering Attacks(Shumailov et al., 2021) to falsify Proof-of-Unlearning (PoUL), with the intent of either enhancing model performance or reducing computational costs.

## 3 ROBUST MACHINE UNLEARNING: PRELIMINARIES

**Notation.** Let $\mathcal{D} = \{(x_i, y_i)_{i=1}^N\}$ denote the training dataset used to train a model $f(\cdot; \theta)$, where $N$ represents the number of training samples, each consisting of input features $x \in \mathbb{R}^d$ and target output $y$, and $\theta$ denotes the model's parameters. The model's training process is represented as $\mathcal{A}(f(\cdot; \theta), \mathcal{D})$, while the unlearning process is denoted by $\mathcal{U}(f(\cdot; \theta), \mathcal{D}_u)$, where $\mathcal{D}_u \subset \mathcal{D}$ is the subset of data to be unlearned. After the unlearning process, the updated model is expressed as $f_u(\cdot; \theta^u)$, i.e., $f_u(\cdot; \theta^u) = \mathcal{U}(f(\cdot; \theta), \mathcal{D}_u)$.

**Definition 1** *(Machine Unlearning (Cao & Yang, 2015).) An unlearning process $\mathcal{U}(f(\cdot; \theta), \mathcal{D}_u)$ aims to find an unlearned model $f_u(\cdot; \theta^u)$ so that it closely aligns with a model trained from scratch on the retain set $\mathcal{D}_r = \mathcal{D}/\mathcal{D}_u$, i.e. $f_u(\cdot; \theta^u) = f_r(\cdot; \theta^r) = \mathcal{A}(f(\cdot; \theta), \mathcal{D}_r)$.*

According to Definition 1, for MUL evaluation, the performance of an unlearning algorithm should closely align with that of a retrained model on the retain set. In practice, especially for large-scale systems like foundation models, where retraining is computationally prohibitive or impractical for information unlearning, a more feasible approach is to ensure its performance has sufficient divergence from the original model on the forget set while maintaining performance on the retain set. For example, a generative unlearned model should no longer be capable of producing undesirable information from the forget set (Warnecke et al., 2021; Fan et al., 2024; Li et al., 2024). That is, empirically, an unlearned model should decorrelate the input $x \in \mathcal{D}_u$ from the original output by a significant margin $\varepsilon_1$, while maintaining close predictions for $x \in \mathcal{D}_r$ for its utility:

$$E_{x \in \mathcal{D}_u}[||f_u(x, \theta^u) - f(x, \theta)||] > \varepsilon_1, E_{x \in \mathcal{D}_r}[||f_u(x, \theta^u) - f(x, \theta)||] \leq \varepsilon_2. \quad (1)$$

Here $E$ represents the statistical mean over a distribution, and $\varepsilon_1$ and $\varepsilon_2$ may vary depending on the task and should align with real-world attack detectability and interpretability of human visual inspection. These empirical observations in (1) underpin the commonly used performance metrics for

evaluating unlearning, typically focusing on performance over the retrain set and the forgetting set. However, such evaluations overlook a critical security gap - the possibility of adversarial recovery of forgotten information by arbitrary or adversarial crafted perturbations (Zhang et al., 2025; Tsai et al., 2024; Han et al., 2024; Pham et al., 2023; Yuan et al., 2024) - and therefore fail to comprehensively assess the true effectiveness of unlearning models. In this study, we formulate this vulnerability of MUL in knowledge regulation and information removal as the following two propositions:

**Proposition 2** *For an unlearned generative system $f_u(\cdot, \theta^u)$ that satisfies conditions in (1), there may exist an adversarial probing $\delta_x \notin \mathcal{D}_u$ s.t. $||f_u(\delta_x, \theta^u) - f(x, \theta)|| < \varepsilon_1, \forall x \in \mathcal{D}_u$.*

**Proposition 3** *For an unlearned discriminative model $f_u(\cdot, \theta^u)$ that satisfies conditions in (1), there may exist a small non-zero $\delta_x \in \mathbb{R}^d$ (i.e. $||\delta_x|| < \epsilon$) s.t. $||f_u(x + \delta_x, \theta^u) - f(x, \theta)|| < \varepsilon_1, \forall x \in \mathcal{D}_u$.*

Proposition 2 considers scenarios where a generative model produces outputs based on various prompts or inputs, and Proposition 3 focuses on discriminative tasks where slight perturbations $\delta_x$ on the data $x$ can bypass the unlearning process and the system's security is compromised. Note, $\delta_x$ in both Propositions is data-specific and model-specific. The extra constraint $||\delta_x|| < \epsilon$ for discriminative models ensures semantic similarity between $x$ and $x + \delta_x$, so that the attack remains meaningful and realistic, aligning with its practical use of these models. By contrast, generative models allow for unconstrained $\delta_x$, as the attack focuses exclusively on the outputs generated from crafted inputs, irrespective of input realism.

Since machine unlearning is inherently tied to data security, unlearning algorithms must be robust and resilient to such malicious recovery attempts.

**Proposition 4** *(Robust Unlearning). A unlearning process, $\mathcal{U}(f(\cdot; \theta), \mathcal{D}_u)$, is considered robust if $\forall x \in \mathcal{D}_u$, $\forall \delta_x \in \mathbb{R}^d$, we have conditions in (1), plus $||f_u(\delta_x, \theta^u) - f(x, \theta)|| > \varepsilon_1$ for generative tasks and $||f_u(x + \delta_x, \theta^u) - f(x, \theta)|| > \varepsilon_1$ for discriminative models (where $||\delta_x|| < \epsilon$).*

Intuitively, Robust Unlearning ensures that the system is incapable of producing the specified information, whether under normal conditions or in the presence of adversarial manipulation. We propose this as a comprehensive and robust standard for defining and evaluating unlearning algorithms in our benchmark. For instance, our empirical experiments on discriminative models in Table 1 show that retraining achieves strong robustness, while those evaluated unlearning algorithms are less robust to adversarial attacks, suggesting these vulnerabilities are intrinsic to the unlearning methods.

## 4 RUB Benchmarking Framework

RUB aims to systematically evaluate the robustness of machine unlearning algorithms against adversarial recovery of forgotten information. To this end, we introduce a unified evaluation protocol, which probes whether unlearned models still retain residual information of the unlearned data. To ensure broad applicability, our benchmark spans three distinct tasks: classification, image-to-image reconstruction, and text-to-image synthesis, each representing a different modality and unlearning objective. For each task, we define tailored adversarial attack strategies and evaluation metrics to quantify the model's susceptibility to post-unlearning information recovery. Note, to evaluate the upper bound of recoverable information from unlearned models, we adopt a white-box, worst-case assumption for the adversary attack in our benchmark, that is, RUB has full access to the original model *before* and *after* the unlearning process, along with knowledge of the specific samples designed for removal.

### 4.1 Unlearning Scenarios/Tasks

To comprehensively evaluate the robustness of unlearning algorithms, RUB spans multiple unlearning scenarios. We categorize these into three representative tasks highlighted by blue, green, and orange in Fig. 1, respectively. Each of these tasks targets a fundamentally different form of information retention and thus presents unique challenges for unlearning.

In **discriminative unlearning**, unlearning typically involves removing the influence of specific training samples or entire classes from a discriminative model. This setting is well-studied in terms

of classification and forms the foundation of many early unlearning algorithms. Our benchmark includes classification unlearning as a canonical use case and evaluates both natural forgetting efficacy and adversarial vulnerability. Specifically, following prior arts (Fan et al., 2024), our classification unlearning evaluation will be performed on 3 datasets: *CIFAR10, CIFAR100, and Tiny-ImageNet*.

**Image-to-Image (I2I) reconstruction unlearning** targets to erase a model's ability to reconstruct specific visual content, while preserving its performance on the retained dataset. To instantiate this task, we follow the prior unlearning study SalUn (Fan et al., 2024) for masked image inpainting, where a generative model is pre-trained to reconstruct missing regions from partially masked inputs. After unlearning the forget data, the model's behavior is then assessed by feeding in masked versions of those forgotten images. If unlearning is successful, the model should fail to accurately reconstruct the masked regions, instead producing uninformative or generic outputs. Any successful recovery of the original content, particularly under adversarial probing, suggests residual memorization and a failure to fully forget. In the I2I scenario, we adopt *ImageNet-1k* as the benchmarking set.

**Text-to-Image (T2I) synthesis unlearning** aims to remove the generative model's ability to produce images corresponding to textual prompts. With the rise of powerful DMs, such safety-driven unlearning has become increasingly important. Usually, the forgetting targets are abstract and semantic, such as high-level concepts embedded in a diffusion model. In this task, RUB leverages the unlearning codebase in UnlearnDiffAtk (Zhang et al., 2024d) for this purpose. Specifically, unlearning algorithms are applied to the pre-trained DM for the provided prompt lists in UnlearnDiffAtk, and we evaluate whether unlearned DMs can still be coerced into generating "forgotten" content.

## 4.2 Evaluation Protocol and Metrics

**Evaluation protocol.** In our benchmark, a strong adversarial evaluation protocol is designed to assess the robustness of unlearning algorithms against malicious recovery of forgotten information. To evaluate the upper bound of recoverable information from unlearned models, we adopt a white-box, worst-case assumption for the adversary, that is, RUB has full access to the original model *before* and *after* the unlearning process, along with knowledge of the specific samples designed for removal. As shown in Fig. 1, for a given unlearning task, we begin with a pretrained model trained on the full dataset (i.e. $D_u \cup D_r$). An unlearning algorithm is then applied to remove the influence of target data, resulting in an unlearned model. To assess adversarial robustness, we craft a probing attack (details in Section 4.4) to actively resurface forgotten information. We then evaluate the post-attack performance on the forgetting set.

**Evaluation metrics.** Since unlearning objectives vary significantly across tasks, there are no universally applicable metrics. As such, we design task-specific evaluation metrics as follows. Please refer to the Appendix for detailed information.

For classification unlearning, we utilize both *Unlearning Accuracy* (UA) and *Membership Inference Attack* (MIA) as the evaluation metrics. It is important to clarify the interpretation of UA. In some prior works, UA has been regarded as the unlearning success rate, where higher values indicate better performance. In contrast, we define UA as the model's raw accuracy on the forget dataset, which should be interpreted as lower values indicating better unlearning. For the MIA evaluation, we adopt a shadow-model-based MIA strategy Shokri et al. (2017) (See details in the Appendix). To address the randomness in MIA for reliable evaluation, we randomly sampled 10 fixed random seeds for executing unlearning and 5 fixed random seeds for training MIA attack models. In total, we have 50 sets of results, and their statistics are reported.

For I2I unlearning, we perform both qualitative and quantitative evaluations. Qualitatively, human perception plays a key role, and the primary question is whether the generated image appears visually similar to the original. Quantitatively, we adopt standard image generation metrics, including the Inception Score (IS) (Salimans et al., 2016) and the Fréchet Inception Distance (FID) (Heusel et al., 2017), to measure output quality. A higher IS and a lower FID indicate better generation performance.

For T2I unlearning, we evaluate robustness using two key indicators: the attack success rate and the number of steps required to achieve a successful attack. These metrics reflect how much forgotten information remains recoverable, providing a quantitative measure of the unlearning model's resilience under adversarial probing. To determine attack success, we follow the protocol from Un-

learnDiffAtk (Zhang et al., 2024d), employing a classifier to assess whether the generated image is recognized as belonging to the forgotten class.

## 4.3 SUPPORTED UNLEARNING ALGORITHMS

Our benchmark contains 15 unlearning algorithms, ranging from MUL baseline approaches to the latest DM-specific unlearning methods, among which 6 are applicable to classification unlearning, 2 to I2I unlearning, and 7 to T2I unlearning. Specifically, our classification unlearning can be achieved by FT (Warnecke et al., 2021), RL (Golatkar et al., 2020), IU (Koh & Liang, 2017; Izzo et al., 2021), l1-sparse (Jia et al., 2023), and SalUn (Fan et al., 2024). The two unlearning methods applicable to I2I unlearning in literature are I2I (Li et al., 2024) and SalUn (Fan et al., 2024). For T2I unlearning, various DM-tailored algorithms, AdvUnlearn (Zhang et al., 2024c), EraseDiff (Wu et al., 2025), ESD (Gandikota et al., 2023), FMN (Zhang et al., 2024b), SalUn (Fan et al., 2024), ScissorHands (Wu & Harandi, 2024), and SPM (Lyu et al., 2024), are evaluated in this benchmark.

## 4.4 ADVERSARIAL RECOVERY ATTACKS

One key module in RUB is the adversarial attack that is able to exploit residual knowledge within the unlearned model for forgotten information resurfacing. This process reveals the extent to which existing unlearning techniques eliminate traces of forgotten data, providing a direct empirical measure of unlearning robustness. Although several adversarial recovery attacks have been proposed for DMs and LLMs, they are not applicable to the canonical discriminative unlearning and I2I unlearning scenarios. To this end, we introduce a generic white-box post-unlearning attack prototype, namely Unlearning Mapping Attack. It should be noted that though UMA is introduced as the default attack module in this benchmark, other attacks can be easily plugged in.

**Definition 5** *(Generic UMA) In the context of MUL, an adversarial strategy is considered UMA given that the attack, by **modifying the input** to a model, causes the output to **reveal or approximate** information or concept that was intended to be removed or forgotten through the unlearning process.*

Notably, UMA probes an unlearning model by varying its input in inference only; It does not require any change to the unlearning algorithm or the model parameters after the unlearning process. Let $D$ quantifies the behavior difference of two models, a white-box UMA can be formulated as

$$\arg \min_{\delta_i} D[f_u(\delta_i, \theta^u), f(x_i, \theta)], \forall x_i \in \mathcal{D}_u. \tag{2}$$

Here, $D$ can vary depending on the context, such as the KL Divergence on classification logits, and the Mean Square Error for I2I reconstruction. The UMA formulation in Eq. (2) aligns with the proposition of robust unlearning. If for every $x \in \mathcal{D}_u$ we find an optimal $\delta_x$ to minimize the difference, and the minimum difference is still larger than $\varepsilon_1$, we can conclude that the unlearned model is robust with respect to $\varepsilon_1$.

Unlike the models in the first two unlearning tasks, which produce outputs in a single forward pass, DMs in T2I unlearning are inherently iterative generative processes. Accordingly, rather than comparing pre-attack and post-attack image outputs for UMA calculation, we can leverage DM's training mechanism, i.e. noise level estimation, as the quantitative performance measurement in Eq. (2). Under this formulation, the UMA attack for T2I tasks is instantiated as follows.

$$\arg \min_{\delta_i} E_t ||\epsilon_{\theta^u}(x_i^t|\delta_i) - \epsilon_\theta(x_i^t|c)||, \forall x_i \in \mathcal{D}_u, \tag{3}$$

where $E_t$ indicates that the attack is optimized over all $t$ steps in the diffusion process, and $c$ and $\delta_i$ are the original and adversarial textual prompts, respectively. It is worth noting that the UMA objective in Eq. (3) closely resembles that of existing adversarial attacks on unlearned diffusion models (Tsai et al., 2024; Zhang et al., 2024d). Therefore, we adopt the gradient-based attack in UnlearnDiffAtk (Zhang et al., 2024d) in our benchmark to recover forgotten information.

| CIFAR10 | No Atk | | | $\epsilon = 8/255$ | | $\epsilon = 16/255$ | |
|---|---|---|---|---|---|---|---|
| | TA ↑ | UA ↓ | MIA ↓ | UA↓ | MIA↓ | UA↓ | MIA↓ |
| Original | 94.13 | 100 | 0.9796 | - | - | - | - |
| retrain | 94.14 | 0 | 0 | 0 | 0 | 0 | 0 |
| FT | 91.82 | 20.55 | 0.088 | 99.96 | 0.995 | 99.98 | 0.995 |
| RL | 92.19 | 0 | 0 | 5.82 | 0.024 | 26.64 | 0.135 |
| IU | 88.06 | 8.76 | 0.058 | 99.12 | 0.965 | 99.87 | 0.983 |
| $l_1$-sparse | 90.00 | 0 | 0 | 98.30 | 0.871 | 99.90 | 0.978 |
| SalUn | 92.70 | 0 | 0 | 7.13 | 0.036 | 26.87 | 0.148 |

| CIFAR100 | No Atk | | | $\epsilon = 8/255$ | | $\epsilon = 16/255$ | |
|---|---|---|---|---|---|---|---|
| | TA↑ | UA↓ | MIA↓ | UA↓ | MIA↓ | UA↓ | MIA↓ |
| Original | 75.25 | 100 | 0.9908 | - | - | - | - |
| retrain | 75.40 | 0 | 0.024 | 0 | 0.014 | 0 | 0.014 |
| FT | 67.64 | 0.48 | 0.306 | 99.28 | 0.977 | 99.89 | 0.992 |
| RL | 69.96 | 3.20 | 0.269 | 51.53 | 0.688 | 80.50 | 0.790 |
| IU | 66.42 | 53.37 | 0.848 | 99.93 | 1 | 99.93 | 1 |
| $l_1$-sparse | 70.70 | 1.30 | 0.402 | 99.77 | 0.925 | 99.91 | 0.945 |
| SalUn | 73.89 | 4.13 | 0.221 | 61.57 | 0.788 | 85.16 | 0.888 |

| Tiny-ImageNet | No Atk | | | $\epsilon = 8/255$ | | $\epsilon = 16/255$ | |
|---|---|---|---|---|---|---|---|
| | TA↑ | UA↓ | MIA↓ | UA↓ | MIA↓ | UA↓ | MIA↓ |
| Original | 64.17 | 99.96 | 1 | - | - | - | - |
| retrain | 57.74 | 0 | 0 | 0 | 0 | 0 | 0 |
| FT | 60.48 | 79.01 | 0.721 | 99.99 | 0.991 | 99.99 | 0.991 |
| RL | 56.23 | 2.09 | 0.028 | 99.78 | 0.859 | 99.99 | 0.917 |
| IU | 57.71 | 94.44 | 0.882 | 99.99 | 0.991 | 99.99 | 0.991 |
| $l_1$-sparse | 58.28 | 45.99 | 0.228 | 99.99 | 0.833 | 99.99 | 0.843 |
| SalUn | 57.82 | 5.95 | 0.084 | 99.98 | 0.964 | 99.99 | 0.982 |

Table 1: Test Accuracy (TA), Unlearning Accuracy (UA), and MIA scores before and after adversarial attacks for the Class Unlearning scenario. Attack is bounded with 8/255 and 16/255. The original here indicates the model performance before unlearning.

## 5 EXPERIMENTS AND DISCUSSIONS

### 5.1 IMPLEMENTATION DETAILS

**Discriminative unlearning.** We choose ResNet50 as our model backbone and conduct class-wise unlearning experiments on all three datasets. For each dataset, we randomly pick 10% of the total classes to evaluate unlearning performance. In our case, class 0 is selected for CIFAR10, class 1, 2, 24, 27, 41, 50, 52, 73, 78, 91 for CIFAR100, and class 3, 11, 17, 34, 49, 59, 97, 107, 109, 129, 133, 137, 154, 156, 173, 177, 179, 183, 194, 197 for Tiny-ImageNet as the unlearning target. Since the attack for discrimination tasks unlearning needs to be bounded, or it would be meaningless otherwise, we bound the perturbation to 8/255 and 16/255, which represent the maximum noise strength while preserving major feature information.

**Image-to-Image unlearning.** We follow the study in I2I attack (Li et al., 2024) and adopt class unlearning using Masked AutoEncoder (MAE) on ImageNet-1k dataset. We utilize the class index file from I2I attack project page as our dataset, which contains 200 class indices from ImageNet-1k. The first 100 classes are selected to be the forget set, while the last 100 classes are the retain set.

**Text-to-Image unlearning.** We evaluate a range of T2I unlearning methods on the task of forgetting specific objects and concepts. Following prior work in T2I unlearning (Zhang et al., 2024d), RUB includes the following target concepts for removal: Church, Garbage Truck, Nudity, Parachute, and Tench. To compute the average number of steps required to compromise unlearning, we cap the maximum number of adversarial prompt optimization iterations at 100.

We also present full details of our implementation, such as unlearning hyperparameters and checkpoints we used for evaluation in the Appendix.

| | IS↑ | | | | | | FID↓ | | | | | |
| | No Atk | | 8/255 | | Unbound | | No Atk | | 8/255 | | Unbound | |
| | R | F | R | F | R | F | R | F | R | F | R | F |
|---|---|---|---|---|---|---|---|---|---|---|---|---|
| Original | 6.21 | 6.39 | - | - | - | - | 96.12 | 103.38 | - | - | - | - |
| I2I | 6.18 | 2.79 | 6.21 | 6.22 | 6.20 | 6.35 | 100.15 | 306.43 | 94.76 | 114.16 | 96.98 | 110.29 |
| SalUn | 6.05 | 2.42 | 6.02 | 6.11 | 6.13 | 6.27 | 130.45 | 330.79 | 102.75 | 133.82 | 94.15 | 108.44 |

Table 2: IS and FID results for image-to-image generation unlearning. *R* and *F* stand for retain set and forget set. The attack strength is set to 0, 8/255, and unbound (where the noise strength is unlimited). Note that a higher IS score indicates better image quality, while a lower FID score reflects improved image fidelity.

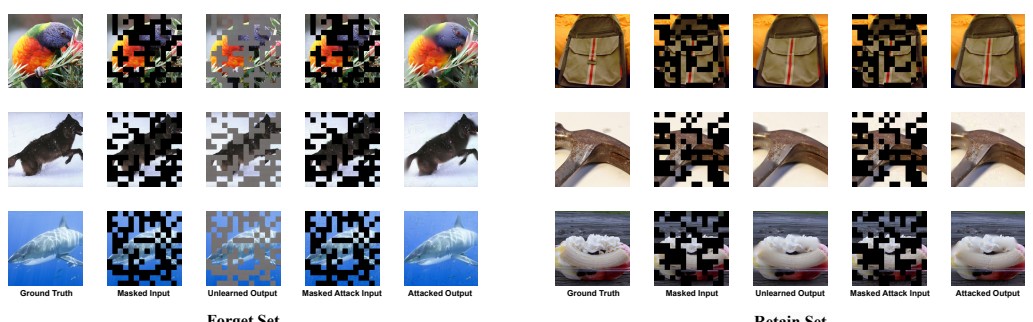

Figure 2: Unlearning Mapping Attack on image generation unlearning. I2I (Li et al., 2024) unlearning method is tested here. Reconstructed images are from ImageNet1k dataset.

## 5.2 EVALUATION RESULTS AND DISCUSSIONS

**Discriminative unlearning.** Table 1 represents our evaluation results on classification unlearning tasks. Generally, class-level unlearning shows varying robustness against unlearning mapping attacks, especially the strong performance of retraining methods. This not only provides empirical evidence that a model can satisfy the robust unlearning criteria but also validates that class-wise discriminative unlearning effectively aligns the unlearned model with a retrained model. However, we notice that as the dataset becomes more complex, from CIFAR10 to Tiny-ImageNet, both unlearning efficacy and robustness drop vastly. While unlearning's golden standard, the retraining method, is still capable of producing a reliable unlearning outcome, other unlearning methods struggle to keep up. Even though some maintain good unlearning efficacy, their robustness drops vastly.

Note that we also include Test Accuracy(TA) in our evaluation metrics. While TA does not indicate any robustness of an unlearning method, it does reflect the strength of the unlearning. Since most unlearning methods have some hyperparameters that can adjust their strength, with stronger unlearning resulting in lower test accuracy, one can perform very strong unlearning to have good unlearning performance (low UA and MIA) and very low test accuracy, which is impractical. Therefore, TA serves as a balancing metric, where lower TA can imply stronger unlearning, but the trade-off needs to be considered for robustness.

**I2I unlearning.** Table 2 and Figure 2, 3 present qualitative and quantitative results for image generation tasks. Even when the unlearned model avoids reconstructing images from the forget set, it can still generate well-restored images when given an adversarial input. Note that the attack can be unbounded for generation tasks since we only care about the model's output. However, as shown in both figures, even small noise is sufficient for a successful attack, exposing the model's vulnerability. Moreover, the unlearning mapping attack does not significantly impact samples from the retain set, indicating that distinguishing between the forget and retain sets in the verification process is unnecessary.

**T2I unlearning.** We represent the results in Table 3. While ASR requires consideration of both pre-attack and post-attack performance, the average attack step provides a clearer and more dis-

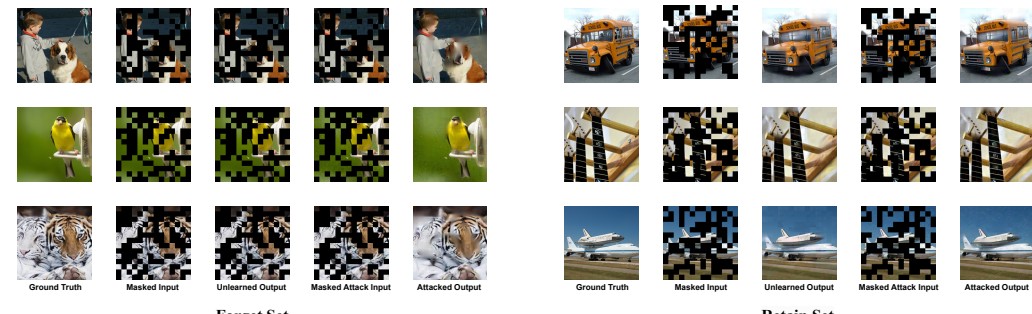

Figure 3: Unlearning Mapping Attack on image generation unlearning. SalUn (Fan et al., 2024) unlearning method is tested here. Reconstructed images are from ImageNet1k dataset.

|              | Church       | Garbage Truck | Nudity      | Parachute   | Tench       |
|--------------|--------------|---------------|-------------|-------------|-------------|
| AdvUnlearn   | 0/0.08       | 0/0.06        | 0.076/0.356 | 0.02/0.14   | 0/0.06      |
|              | 93.94/100    | 95.82/100     | 72.91/100   | 89.18/100   | 95.54/100   |
| EraseDiff    | 0.06/0.76    | 0.06/0.52     | 0/0.059     | 0.04/0.82   | 0/0.12      |
|              | 45.72/38     | 64.68/88.5    | 95.37/100   | 29.44/8.5   | 92.94/100   |
| ESD          | 0.14/0.76    | 0.02/0.44     | 0.212/0.907 | 0.06/0.8    | 0/0.46      |
|              | 38.58/20.5   | 73.98/100     | 25.86/11    | 44.46/33    | 67.04/100   |
| FMN          | 0.52/0.96    | 0.46/0.98     | 0.881/1     | 0.52/1      | 0.36/1      |
|              | 6.78/0       | 3.96/1        | 0.39/0      | 2.1/0       | 6.42/2.5    |
| SalUn        | 0.1/0.68     | 0.02/0.5      | 0.017/0.246 | 0.08/0.88   | 0/0.24      |
|              | 47.16/33     | 67.72/93      | 84.71/100   | 30.32/14    | 88.4/100    |
| Scissorhands | 0/0.08       | 0/0.04        | 0/0.119     | 0.02/0.44   | 0/0.18      |
|              | 96.36/100    | 99.7/100      | 92.35/100   | 78.38/100   | 91.58/100   |
| SPM          | 0.44/0.96    | 0.04/0.82     | 0.559/1     | 0.26/0.96   | 0.06/0.94   |
|              | 7.52/1       | 37.36/23      | 1.69/0      | 12.34/3     | 17.92/8.5   |

Table 3: Empirical experiment on Text-to-Image unlearning methods. Five objects or concepts are selected to test unlearning robustness. The results recorded in each cell are **pre/post ASR** (first row) and **average/median attack steps** (second row).

tinct separation between methods. These results indicate that although many unlearning approaches demonstrate strong efficacy, their robustness against adversarial attacks remains limited. It is also worth noting that for both Image-to-Image and Text-to-Image unlearning, the retraining baseline is absent, as retraining is typically impractical due to the scale of the training data.

**Overall**, our results across discriminative, Image-to-Image, and Text-to-Image unlearnings reveal a consistent gap: while many unlearning methods achieve reasonable efficacy, they pay little attention to robustness against adversarial attacks. Consequently, existing approaches exhibit fundamental weaknesses once probed with adversarial strategies. This observation directly supports the motivation outlined in Section 3.2 and Definition 4, highlighting the necessity of Robust Unlearning as a core requirement rather than an optional property. We hope that future research will explicitly incorporate robustness considerations into the design of unlearning algorithms, moving toward solutions that are not only effective but also resilient against adversarial recovery.

## 6 CONCLUSION

We present RUB, the first comprehensive benchmark designed to evaluate unlearning algorithms under adversarial recovery attacks. RUB spans three vision tasks and supports standardized evaluation protocols, enabling a systematic comparison of unlearning robustness. Our proposed UMA formulation and its task-specific implementations allow empirical probing of residual knowledge in unlearned models. Extensive experiments across 15 representative methods reveal that current unlearning approaches often fail to fully erase sensitive information. We hope RUB provides a rigorous foundation for future research into robust, attack-resilient machine unlearning.

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
