## A UMA IMPLEMENTATION AND ABLATIONS

### A.1 GRADIENT-BASED IMPLEMENTATION OF UMA

To achieve the attack goal outlined in (2), we introduce a gradient-based input mapping attack:

$$\arg\min_{\{\delta_x\}} E_{x \in \mathcal{D}_u} \left[ D(f_u(\delta_x; \theta^u), f(x; \theta)) \right],\tag{4}$$

where $D$ quantifies the difference and can vary depending on the context, such as Mean Square Error Loss, Binary Cross-Entropy Loss, or KL Divergence Loss. To solve this optimization problem, we adopt the Projected Gradient Descent (PGD) method in Madry et al. (2017) to find the input $\delta$. While PGD is normally used to maximize empirical loss, in this case, we aim to *minimize* the loss, thus taking the opposite direction of the gradient update:

$$\delta_x^{t+1} = \delta_x^t - \alpha \cdot sign[\nabla_{\delta_x} D(f_u(\delta_x^t; \theta^u), f(x; \theta)],\tag{5}$$

where $\alpha$ stands for the step size for each iteration. The pseudocode of UMA is provided in Algorithm 1. For simplicity, we only adopt the PGD-based mapping method as our baseline, though other optimization techniques can be substituted for potentially better performance.

Though UMA and Robust Unlearning are consistent in principle, the optimization problem is typically non-convex. As a result, Algorithm 1 does not guarantee exploration of all possible perturbations. Yet, it still provides a practical and actionable framework for identifying vulnerabilities in unlearning methods. Even in cases where UMA does not succeed, the absence of successful attacks strengthens the empirical evidence that the model may satisfy the robust unlearning criteria.

---

**Algorithm 1** Unlearning Mapping Attack

1: **Input:** Pre-trained model $f(\cdot; \theta)$, Unlearned model $f_u(\cdot; \theta^u)$, Unlearning dataset $\mathcal{D}_u$, Attack steps $T$, Attack step size $\eta$
2: **Output:** Attack dataset $\mathcal{D}_{atk}$
3: Random initialize attack noise $\{\delta_x\}$ for $x \in \mathcal{D}_u$
4: **for** $k = 0$ **to** $T$ **do**
5:     Calculate loss $\psi \leftarrow \sum_{x \in \mathcal{D}_u} D(f(x; \theta), f_u(\delta_x^k; \theta^u))$
6:     Update attack noise $\{\delta_x^{k+1}\} \leftarrow \{\delta_x^k\} - \eta \cdot sign(\nabla_{\delta_x} \psi)$
7:     $\{\delta_x^{k+1}\} \leftarrow clip(\{\delta_x^{k+1}\}, 0, 1)$
8: **end for**
9: Construct attack dataset $\mathcal{D}_{atk} \leftarrow (\delta_x^{k+1}, y_x)$

---

### A.2 ABLATION STUDY

We conduct ablation experiments on the two hyperparameters in UMA, the number of steps and step size. All experiments are done using discriminative models on CIFAR10 dataset. SalUn (Fan et al., 2024) is chosen as the unlearning algorithm. All ablation experiments on step sizes have a fixed number of steps of 100, and all ablations on iteration numbers have a fixed step size of 1/255. Attack strength is set to 16/255 across all ablations.

As shown in Figure 4, the attack efficacy generally increases as the number of steps goes up. However, higher iteration numbers result in greater computation costs, which form a trade-off that the attacker needs to make. On the other hand, as shown in Figure 5, the attack step size reaches its best performance, around 0.7/255 to 1/255. A larger step size will cause the attack to find an incorrect direction, reducing the attack efficacy, while a smaller step size will generally cause a slow convergence speed, requiring a larger iteration step to reach equivalent performance.

## B DETAILS ON I2I UNLEARNING SETUP

In the experiments on image-to-image generative unlearning models, we evaluate whether our UMA attacks could explore the residue information left in the model after unlearning and resurface the "forgotten" knowledge. To this end, we follow the previous arts in I2I where the generative model is used to recover the masked region in a query image. To ease the discussion, let's first clarify the data

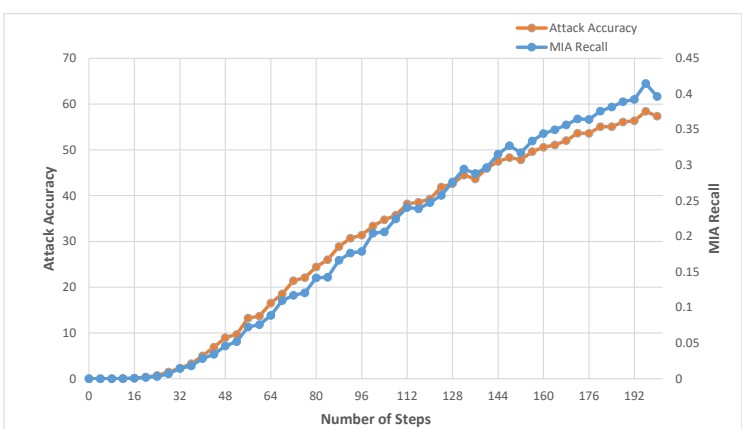

Figure 4: Ablation on attack iteration numbers. The experiments are done on CIFAR10 using SalUn (Fan et al., 2024) as the baseline unlearning algorithm. All experiments have a fixed step size of 1/255 and an attack strength of 16/255.

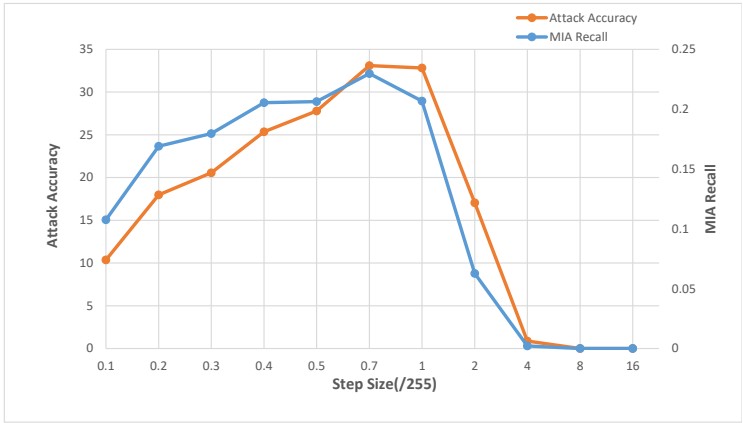

Figure 5: Ablation on attack step size. The experiments are done on CIFAR10 using SalUn (Fan et al., 2024) as the baseline unlearning algorithm. All experiments have a fixed number of steps of 100 and an attack strength of 16/255.

| L1 per image | ISI (Li et al., 2024) | | SalUn (Fan et al., 2024) | |
|---|---|---|---|---|
| | No Attack | 8/255 | No Attack | 8/255 |
| Retain set | 64,619 | 42,410 | 214,596 | 114,089 |
| Forget set | 1,140,778 | 48,317 | 2,790,552 | 242,029 |

Table 4: L1 norm between the outputs of the generative model before and after unlearning. The values under no attack are calculated by $L1(I_2, I_1)$, and the values under the attack strength 8/255 are computed by $L1(I_3, I_1)$.

flow and pipeline of the generative model experiment. In our experiments, the generative unlearning pipeline involves the following steps:

- $I_0$: The ground truth image from the forget set.
- $I_m$: The masked version of the image $I_0$, which serves as the input to the generative model.
- $I_1$: The output of the original generative model (before unlearning), where the masked regions in $I_m$ are reconstructed.
- $I_2$: The output of the unlearned generative model, which cannot reconstruct the masked regions for the forget set and instead generates gray or noisy outputs.
- $I_3$: The output of the unlearned generative model when attacked with UMA, which aims to resurface the forgotten information and reconstruct the masked regions as $I_1$.

By design, $I_1$, $I_2$, and $I_3$ are naturally different from the masked input $I_m$, as the goal of the generative model is to reconstruct the missing regions. Additionally, for the forget set, $I_2$ differs significantly from $I_1$, as the unlearned model is intended to "forget" the knowledge and cannot recover $I_0$ from $I_m$. UMA's goal is to probe whether the unlearned model can generate $I_3$ that closely resembles $I_1$, thereby bypassing the unlearning mechanism. Based on the above context, UMA's efficacy is evaluated by how closely $I_3$ (the UMA output) resembles $I_1$ (the output of the original generative model before unlearning). This indicates whether the unlearned model retains residual knowledge of the forget set, effectively failing to fully "forget."

To verify UMA's impact, we directly computed the L1 distance between $I_3$ and $I_1$ per image. As shown in the Table 4, the L1 differences between $I_1$ and $I_3$ are very small after the attack (e.g. for the 224x224x3 image, average 0.3 intensity difference per pixel for the forget set with I2I (Li et al., 2024) and 1.6 intensity difference per pixel for the SalUn (Fan et al., 2024)), indicating that UMA can prompt the unlearned model to output information it was supposed to forget. This provides strong evidence that UMA effectively bypasses the unlearning process.

In addition, we include multiple visual examples in Figure 6 and 7. These examples present images for $I_0$, $I_m$, $I_1$, $I_2$, and $I_3$, providing a clear comparison of the reconstruction results across all stages of the pipeline. These visualizations demonstrate how UMA successfully recovers information that should have been forgotten, illustrating its effectiveness in attacking the unlearning mechanism.

## C  EXPERIMENTAL EVALUATION ON DISCRIMINATIVE UNLEARNING

### C.1  MIA IMPLEMENTATION

For the MIA evaluation of discriminative unlearning, we adopt a shadow-model-based MIA strategy Shokri et al. (2017) for the quantitative measurement. Specifically, 10% of the total dataset is randomly sampled to train 10 shadow models, each implemented as a ResNet50 and trained for 10 epochs(20 epochs for Tiny-ImageNet). We then collect the logit outputs of these shadow models on both their seen and unseen data to construct the shadow dataset. Using this dataset, we train simple attack models designed to determine whether a given logit is from seen or unseen data. To ensure fine-grained discrimination, we employ one independent attack model per class. Finally, the attack recall is recorded and reported. To address the randomness in MIA for reliable evaluation, we randomly sampled 10 fixed random seeds for executing unlearning and 5 fixed random seeds for training MIA attack models. In total, we have 50 sets of results, and their average and standard deviation are reported in Table 5.

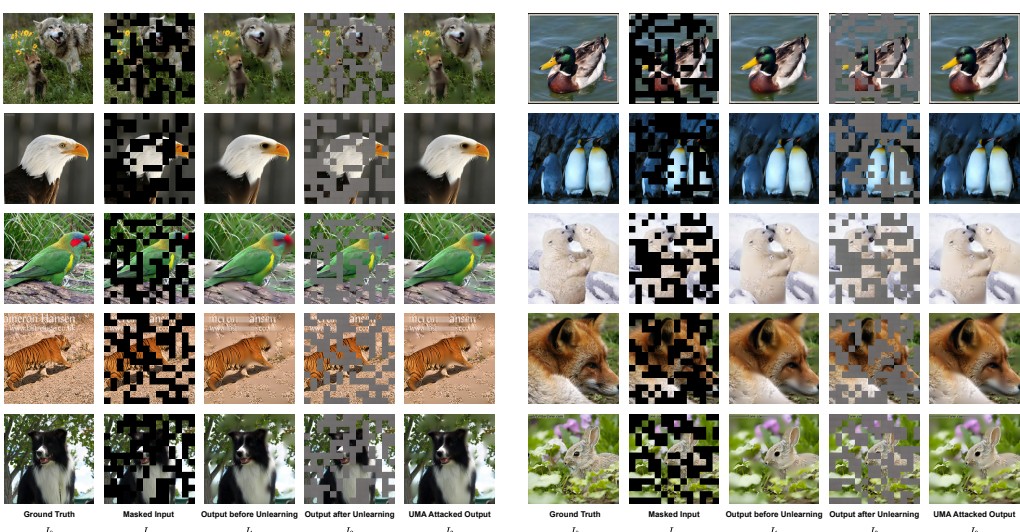

Figure 6: Examples of the generated images using I2I (Li et al., 2024) unlearning methods. Ground truth, $I_0$, Masked Input, $I_m$, Output before Unlearning, $I_1$, Output after Unlearning, $I_2$, UMA Attacked Output, $I_3$, are represented here as discussed in Section A.3

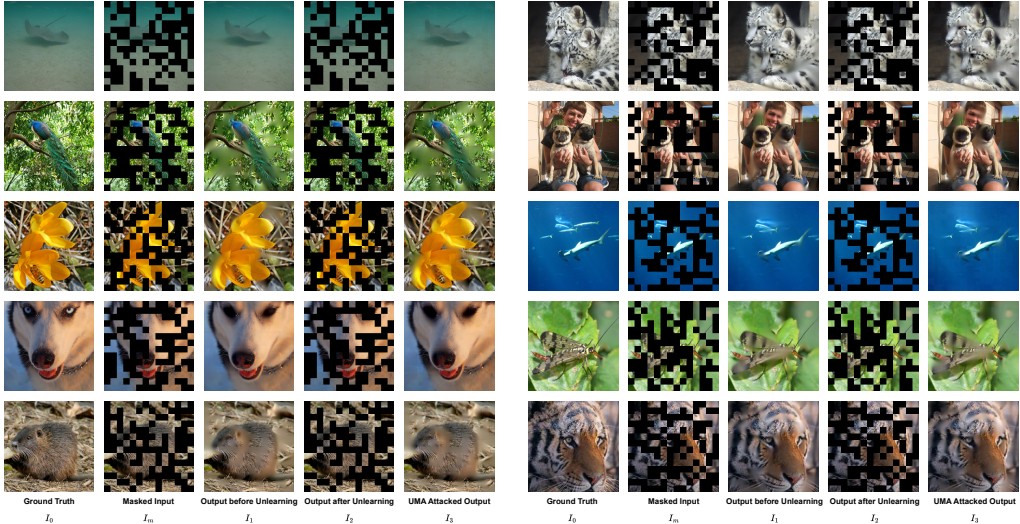

Figure 7: Examples of the generated images using SalUn (Fan et al., 2024) unlearning methods. Ground truth, $I_0$, Masked Input, $I_m$, Output before Unlearning, $I_1$, Output after Unlearning, $I_2$, UMA Attacked Output, $I_3$, are represented here as discussed in Section A.3

| CIFAR10 | No Atk | | | $\epsilon = 8/255$ | | $\epsilon = 16/255$ | |
|---|---|---|---|---|---|---|---|
| | TA↑ | UA↓ | MIA↓ | UA↓ | MIA↓ | UA↓ | MIA↓ |
| Original | 94.13 | 100 | 0.9796 | - | - | - | - |
| retrain | $94.14_{\pm0.20}$ | $0_{\pm0}$ | $0_{\pm0}$ | $0_{\pm0}$ | $0_{\pm0}$ | $0_{\pm0}$ | $0_{\pm0}$ |
| FT | $91.82_{\pm0.42}$ | $20.55_{\pm4.18}$ | $0.088_{\pm0.029}$ | $99.96_{\pm0.03}$ | $0.995_{\pm0.034}$ | $99.98_{\pm0.02}$ | $0.995_{\pm0.003}$ |
| RL | $92.19_{\pm0.43}$ | $0_{\pm0}$ | $0_{\pm0}$ | $5.82_{\pm5.23}$ | $0.024_{\pm0.024}$ | $26.64_{\pm18.28}$ | $0.135_{\pm0.111}$ |
| IU | $88.06_{\pm2.51}$ | $8.76_{\pm5.70}$ | $0.058_{\pm0.040}$ | $99.12_{\pm0.38}$ | $0.965_{\pm0.012}$ | $99.87_{\pm0.09}$ | $0.983_{\pm0.008}$ |
| $l_1$-sparse | $90.00_{\pm0.16}$ | $0_{\pm0.01}$ | $0_{\pm0}$ | $98.30_{\pm0.41}$ | $0.871_{\pm0.081}$ | $99.90_{\pm0.04}$ | $0.978_{\pm0.018}$ |
| SalUn | $92.70_{\pm0.25}$ | $0_{\pm0}$ | $0_{\pm0}$ | $7.13_{\pm9.59}$ | $0.036_{\pm0.062}$ | $26.87_{\pm16.39}$ | $0.148_{\pm0.150}$ |

| CIFAR100 | No Atk | | | $\epsilon = 8/255$ | | $\epsilon = 16/255$ | |
|---|---|---|---|---|---|---|---|
| | TA↑ | UA↓ | MIA↓ | UA↓ | MIA↓ | UA↓ | MIA↓ |
| Original | 75.25 | 100 | 0.9908 | - | - | - | - |
| retrain | $75.40_{\pm1.04}$ | $0_{\pm0}$ | $0.024_{\pm0.015}$ | $0_{\pm0}$ | $0.014_{\pm0.010}$ | $0_{\pm0}$ | $0.014_{\pm0.011}$ |
| FT | $67.64_{\pm1.22}$ | $0.48_{\pm0.26}$ | $0.306_{\pm0.060}$ | $99.28_{\pm0.17}$ | $0.977_{\pm0.031}$ | $99.89_{\pm0.03}$ | $0.992_{\pm0.015}$ |
| RL | $69.96_{\pm0.51}$ | $3.20_{\pm2.88}$ | $0.269_{\pm0.060}$ | $51.53_{\pm3.62}$ | $0.688_{\pm0.080}$ | $80.50_{\pm3.08}$ | $0.790_{\pm0.074}$ |
| IU | $66.42_{\pm2.47}$ | $53.37_{\pm7.11}$ | $0.848_{\pm0.054}$ | $99.93_{\pm0.03}$ | $1_{\pm0}$ | $99.93_{\pm0.02}$ | $1_{\pm0}$ |
| $l_1$-sparse | $70.70_{\pm0.61}$ | $1.30_{\pm0.25}$ | $0.402_{\pm0.099}$ | $99.77_{\pm0.05}$ | $0.925_{\pm0.056}$ | $99.91_{\pm0.03}$ | $0.945_{\pm0.049}$ |
| SalUn | $73.89_{\pm0.34}$ | $4.13_{\pm3.55}$ | $0.221_{\pm0.038}$ | $61.57_{\pm5.04}$ | $0.788_{\pm0.045}$ | $85.16_{\pm3.43}$ | $0.888_{\pm0.035}$ |

| Tiny-ImageNet | No Atk | | | $\epsilon = 8/255$ | | $\epsilon = 16/255$ | |
|---|---|---|---|---|---|---|---|
| | TA↑ | UA↓ | MIA↓ | UA↓ | MIA↓ | UA↓ | MIA↓ |
| Original | 64.17 | 99.96 | 1 | - | - | - | - |
| retrain | $57.74_{\pm0.67}$ | $0_{\pm0}$ | $0_{\pm0}$ | $0_{\pm0}$ | $0_{\pm0}$ | $0_{\pm0}$ | $0_{\pm0}$ |
| FT | $60.48_{\pm0.19}$ | $79.01_{\pm0.69}$ | $0.721_{\pm0.014}$ | $99.99_{\pm0.01}$ | $0.991_{\pm0.004}$ | $99.99_{\pm0.01}$ | $0.991_{\pm0.003}$ |
| RL | $56.23_{\pm0.31}$ | $2.09_{\pm0.31}$ | $0.028_{\pm0.009}$ | $99.78_{\pm0.11}$ | $0.859_{\pm0.023}$ | $99.99_{\pm0.01}$ | $0.917_{\pm0.028}$ |
| IU | $57.71_{\pm1.82}$ | $94.44_{\pm4.26}$ | $0.882_{\pm0.052}$ | $99.99_{\pm0.01}$ | $0.991_{\pm0.004}$ | $99.99_{\pm0.01}$ | $0.991_{\pm0.003}$ |
| $l_1$-sparse | $58.28_{\pm0.35}$ | $45.99_{\pm0.67}$ | $0.228_{\pm0.028}$ | $99.99_{\pm0.01}$ | $0.833_{\pm0.017}$ | $99.99_{\pm0.01}$ | $0.843_{\pm0.019}$ |
| SalUn | $57.82_{\pm0.15}$ | $5.95_{\pm0.78}$ | $0.084_{\pm0.019}$ | $99.98_{\pm0.01}$ | $0.964_{\pm0.017}$ | $99.99_{\pm0.01}$ | $0.982_{\pm0.010}$ |

Table 5: Full evaluation of Test Accuracy (TA), Unlearning Accuracy (UA), and MIA scores before and after Unlearning Mapping Attack for the Class Unlearning scenario. Attack is bounded with 8/255 and 16/255. The original here indicates the model performance before unlearning.

## C.2 INSTANCE-LEVEL UNLEARNING RESULTS

For instance-level classification unlearning, as shown in Table 6, all baseline methods display limited robustness against unlearning mapping attacks. While the retraining method performs the best, it still lacks sufficient robustness, even with $\epsilon = 8/255$. This suggests that attackers can easily manipulate unlearned images, causing the model to re-recognize them, thus compromising the unlearning process.

## D HYPERPARAMETER SETTINGS OF OUR PRE-TRAINED UNLEARNINGS

We perform the **Discriminative Unlearning** and **Image-to-Image Unlearning** by ourselves. Specifically, we adopt the discriminative unlearning code from SalUn's project, https://github.com/OPTML-Group/Unlearn-Saliency, and we use code from I2I (Li et al., 2024), https://github.com/jpmorganchase/i2i_mage/tree/i2i, as reference when constructing Image-to-Image unlearning evaluations, as well as the unlearning class index when performing Image-to-Image Unlearning. We list our detailed hyperparameter selection for discriminative unlearning in Table 7. For Image-to-Image Unlearning, we use the SGD optimizer, learning rate 0.01 for Salun, and the AdamW optimizer, base learning rate 1e-4 for I2I. For the **Text-to-Image** task, we utilize the unlearned model checkpoint from UnlearnDiffAtk (Zhang et al., 2024d)'s project page, https://github.com/OPTML-Group/Diffusion-MU-Attack, for evaluation.

| CIFAR10 | No Atk | | | 8/255 | | 16/255 | |
|---|---|---|---|---|---|---|---|
| | TA | UA | MIA | UA | MIA | UA | MIA |
| Original | 94.13 | 100 | 0.9732 | - | - | - | - |
| retrain | 93.34 | 93.78 | 0.8636 | 99.98 | 0.9774 | 99.98 | 0.9728 |
| FT | 92.13 | 98.02 | 0.9124 | 99.98 | 0.9794 | 99.96 | 0.9810 |
| RL | 89.22 | 91.88 | 0.8012 | 99.96 | 0.9896 | 100 | 0.9866 |
| IU | 89.82 | 97.92 | 0.8926 | 99.98 | 0.9630 | 99.98 | 0.9628 |
| $l_1$-sparse | 91.32 | 95.76 | 0.8848 | 99.98 | 0.9842 | 100 | 0.9814 |
| SalUn | 90.55 | 93.48 | 0.8140 | 100 | 0.9884 | 99.98 | 0.9872 |

| CIFAR100 | No Atk | | | 8/255 | | 16/255 | |
|---|---|---|---|---|---|---|---|
| | TA | UA | MIA | UA | MIA | UA | MIA |
| Original | 75.25 | 100 | 0.9924 | - | - | - | - |
| retrain | 73.92 | 72.72 | 0.7354 | 99.92 | 0.9910 | 100 | 0.9932 |
| FT | 70.90 | 96.44 | 0.9436 | 99.96 | 0.9970 | 100 | 0.9972 |
| RL | 71.05 | 86.04 | 0.7786 | 99.98 | 0.9946 | 100 | 0.9956 |
| IU | 71.89 | 99.20 | 0.9702 | 100 | 0.9894 | 100 | 0.9912 |
| $l_1$-sparse | 69.60 | 90.10 | 0.7404 | 99.98 | 0.9704 | 99.98 | 0.9756 |
| SalUn | 71.99 | 88.72 | 0.7936 | 99.94 | 0.9890 | 100 | 0.9914 |

| Tiny-ImageNet | No Atk | | | 8/255 | | 16/255 | |
|---|---|---|---|---|---|---|---|
| | TA | UA | MIA | UA | MIA | UA | MIA |
| Original | 64.17 | 99.98 | 0.9978 | - | - | - | - |
| retrain | 61.81 | 60.17 | 0.6387 | 99.97 | 0.9735 | 100 | 0.9811 |
| FT | 55.66 | 85.42 | 0.8908 | 99.99 | 0.9969 | 99.97 | 0.9969 |
| RL | 55.36 | 72.88 | 0.8002 | 99.99 | 0.9962 | 99.98 | 0.9968 |
| IU | 56.33 | 94.85 | 0.9591 | 99.97 | 0.9967 | 99.98 | 0.9969 |
| $l_1$-sparse | 56.04 | 61.71 | 0.3836 | 99.99 | 0.7597 | 100 | 0.7614 |
| SalUn | 54.94 | 66.99 | 0.6237 | 99.99 | 0.9654 | 99.99 | 0.9681 |

Table 6: Test Accuracy (TA), Unlearning Accuracy (UA), and MIA scores before and after Unlearning Mapping Attack for the Instance Unlearning scenario. Attack is bounded with 8/255 and 16/255. The original here indicates the model performance before unlearning.

| | CIFAR10 | CIFAR100 | Tiny-ImageNet |
|---|---|---|---|
| FT | epoch=10
lr=0.013 | epoch=10
lr=0.013 | epoch=10
lr=0.0023 |
| RL | epoch=10
lr=0.013 | epoch=10
lr=0.02 | epoch=10
lr=0.0025 |
| IU | $\alpha = 20$ | $\alpha = 20$ | $\alpha = 10$ |
| $l_1$-sparse | epoch=10
lr=0.001
$\alpha = 0.001$ | epoch=10
lr=0.001
$\alpha = 0.0007$ | epoch=10
lr=0.001
$\alpha = 0.0001$ |
| SalUn | epoch=10
lr=0.013 | epoch=10
lr=0.022 | epoch=10
lr=0.0015 |

Table 7: Detailed hyperparameters used for discriminative unlearning evaluations.