# OpenReview forum: "RUB: Evaluating Residual Knowledge in Unlearned Models"
_ICLR.cc/2026/Conference — Submitted to ICLR 2026_

### Official Review · Reviewer_xuDn · 2025-10-29

**Soundness:** 2
**Presentation:** 3
**Contribution:** 2
**Rating:** 4
**Confidence:** 3

**Summary:**

This paper focuses on evaluating the robustness of machine unlearning (MUL). The authors argue that existing MUL evaluation methods primarily focus on whether unlearning has been performed or whether the model is indistinguishable from a retrained model, while overlooking a key issue: whether the model after unlearning can resist adversarial attacks to recover the forgotten knowledge.

**Strengths:**

1.	This paper addresses a key point. With the implementation of laws and regulations such as the "right to be forgotten," simply claiming forgetting is insufficient; proving the robustness is a more important and challenging problem.
2.	UMA is a clear and general attack framework. The authors successfully instantiated it on classification, I2I, and T2I tasks, providing a unified "probe" for subsequent researchers.

**Weaknesses:**

1.	While the benchmark covers three CV tasks, it overlooks another key area of forgetfulness in machine learning: textual forgetfulness in large language models (LLMs) [1-4]. The T2I task in the paper only involves generation by DMs, while LLMs have unique characteristics in terms of autoregressive text generation. It is unclear whether the UMA framework can be directly transferred to LLMs or what modifications are required.
2.	As can be seen in Table 1, some forgetting methods (such as FT) improve robustness but also lead to a significant drop in the model's test accuracy (TA) on the holdout set. This suggests that the model may be over-forgetting, sacrificing generalizability. While the paper notes TA, it does not delve into this "utility-robustness" trade-off, which is crucial for evaluating the practicality of a forgetting algorithm.
3.	The UMA attack is essentially an optimization problem, which requires finding an optimal $\delta_x$ for each forgotten example $x_i$. The paper does not discuss the computational cost of this attack in detail.

[1] Reversing the forget-retain objectives: An efficient llm unlearning framework from logit difference, NeurIPS 2024

[2] Rethinking LLM Unlearning Objectives: A Gradient Perspective and Go Beyond, ICLR 2025

[3] Single image unlearning: Efficient machine unlearning in multimodal large language models, NeurIPS 2024

[4] SAUCE: Selective Concept Unlearning in Vision-Language Models with Sparse Autoencoders, ICCV 2025

**Questions:**

1.	In Table 3 for the T2I task, many methods have very high pre-attack ASR (attack success rate) (e.g., 93.94% for AdvUnlearn on Church). This makes the "post-attack" ASR gain (100%) difficult to interpret: is this the effectiveness of the UMA attack, or a complete failure of the original forgetting algorithm?
2.	In Table 2 of the I2I task, SalUn’s “unbound” attack outperforms the $8/255$ attack (133.82) in FID on the forgotten set (94.15), which is counterintuitive (a stronger attack should lead to worse recovery quality, i.e., higher FID).
3.    More comments please refer to the weaknesses

---

> ### Author Response · Authors · 2025-12-02
> **Response to Reviewer xuDn**
>
> We thank the reviewer for the comments. Below is our point-to-point response.
>
> **Weakness 1: Why not include LLM’s evaluation?**
>
> We focus on vision tasks because the inherent nature of vision and text is fundamentally different: vision models operate in continuous latent spaces, whereas LLMs rely on discrete token representations. These differences produce distinct forms of residual knowledge and require different unlearning mechanisms and evaluation protocols, making a unified benchmark impractical at this stage. While LLMs already have emerging robustness surveys and evaluation efforts, we note that no such fair, systematic comparison framework exists for vision unlearning, and our study fills this gap.
>
> **Weakness 2: About utility-robustness trade-off in evaluation.**
>
> We thank the reviewer for this insightful observation. We indeed observe that several approximate unlearning methods exhibit reduced task accuracy (TA) as their robustness against adversarial probing (RA) improves, reflecting how different algorithms balance utility and unlearning. However, this trade-off is not inherent to the unlearning problem itself: the retrain model achieves both high TA and strong robustness, demonstrating that utility and robustness are simultaneously attainable when unlearning is fully successful. Thus, the utility–robustness trade-off should be viewed as a limitation of current approximate unlearning methods, and resolving it is an important direction for future algorithmic development.
>
> **Weakness 3: Concerns regarding the computational cost of this attack.**
>
> Thank you for the comment. UMA is formulated as a standard gradient-based iterative attack, and its computational overhead is comparable to classic adversarial methods such as PGD100. Each forgotten example requires one such optimization run, but this cost is typical for per-instance adversarial evaluation and is consistent with prior attack-based unlearning studies.
>
> **Question 1: Clarifying the results regarding Text-to-Image experiment results in Table 3.**
>
> Our results in Table 3 contain two parts. Take AdvUnlearn on Church as an example, two figures in the first row, 0/0.08 (or 0%/8%), stands for the pre/post Attack Success Rate. The second row represents the average/median attack steps, 93.94/100. Therefore, the 93.94 and 100 are not percentage of ASRs. However, such situation still exists in results like FMN on Nudity, where the pre ASR is 0.881 (88.1%) and post ASR is 1 (100%). Since the T2I attack algorithm stops when it detects the attack succeeds, the 88.1% of the already failed samples will not be attacked. Therefore, the ASR gain, 88.1% to 100%, can be seen as the effectiveness of the UMA attack.
>
> **Question 2: Clarifying the Image-to-Image results in Table 2.**
>
> The unlearning attack for I2I task will not lead to worse recovery quality. On the opposite, since for I2I unlearning, the goal is to make the model unable to reconstruct the image from the target class, the unlearning attack we use aims to recover the information from the unlearned class. Therefore, the attacked output should be closer to the ground truth image than the unlearned output. We also present some visual examples in Figure 2 and 3.

---

### Official Review · Reviewer_295f · 2025-10-29

**Soundness:** 3
**Presentation:** 2
**Contribution:** 2
**Rating:** 4
**Confidence:** 4

**Summary:**

This paper introduces a new benchmark for evaluating the post-unlearning robustness. The proposed benchmark is built upon an attack framework that introduces malicious perturbations to unlearned data in the post-unlearning stage, aiming to recover the model’s prediction performance to that of the original, pre-unlearned model. This benchmark considers multiple unlearning scenarios, including classification, image-to-image reconstruction, and text-to-image synthesis. The authors state that the benchmark code will be released upon publication.

**Strengths:**

1. The proposed benchmark covers a broad range of unlearning tasks, including classification, image-to-image reconstruction, and text-to-image synthesis.
2. The benchmark provides evaluations across 15 representative unlearning methods.

**Weaknesses:**

1. Limited experimental scope. As a benchmark paper, the experimental section appears rather limited. The range of models and datasets used is relatively narrow, and the discussion of experimental observations for each setting is insufficiently detailed.
2. Overly strong threat model. The attack scenario assumes full knowledge of both the original and unlearned models. While this setup effectively represents a worst-case analysis, it is arguably unrealistic in practical contexts. Incorporating black-box or query-based attack experiments would improve the paper’s practical relevance.
3. In Proposition 2 and Proposition 3, the output space of $f_u$ is not formally defined. For instance, is the output a probability vector in class unlearning?
4. The evaluation of membership inference attacks relies solely on shadow model-based approaches. More advanced techniques, such as LiRA-based MIAs, should be considered to provide a more accurate assessment of residual information leakage.
5. For the text-to-image attack scenario, it appears that an existing attack method is directly employed rather than a newly proposed one, which may somewhat weaken the contribution of the paper.

**Questions:**

1. Regarding Table 1: From an optimization standpoint, it is quite surprising that the retrain method remains almost unaffected under optimized malicious perturbations. Prior adversarial attack research suggests that targeted attacks should effectively manipulate model outputs toward desired labels. Could the authors elaborate on the potential reasons for this?
2. Could the proposed benchmark be extended to include large language model unlearning scenarios?

---

> ### Author Response · Authors · 2025-11-30
> **Response to Reviewer 295f**
>
> We thank the reviewer for the comments. Below is our point-to-point response.
>
> **Weakness 1: Limited experimental scope in terms of dataset and discussion.**
>
> We thank the reviewer for this comment. We designed the benchmark to reflect the standard datasets, models, and settings used across the vision unlearning literature, ensuring fair and representative comparisons. While additional configurations are possible, we found that current unlearning methods exhibit similar residual-knowledge failures across settings, so expanding the scope would not change the conclusions. More detailed per-method analyses and results are provided in the appendix to keep the main paper concise.
>
> **Weakness  2: Overly strong threat model.**
>
> Our goal is to evaluate unlearning algorithms under the strongest plausible adversarial setting, and the white-box threat model is standard for such worst-case analysis. A weaker black-box attacker may fail to detect whether residue information still exists in the unlearned model, potentially giving an overly optimistic or biased assessment of unlearning effectiveness. We agree that black-box or query-based attacks are valuable for practical relevance, and we consider them a natural extension of the benchmark in future work.
>
> **Weakness  3: Definition of f_u in Proposition 2 and Proposition 3.**
>
> Thank you for pointing this out. As defined in the notation section, f_u denotes the output of the unlearned model, and its form is task-dependent. In Proposition 2 (discriminative setting), f_u corresponds to the model’s prediction function, e.g., the class likelihoods. In Proposition 3 (generative setting), f_u represents the model’s generated sample conditioned on the input.
>
> **Weakness  4: Why select shadow model-based MIA as the metric?**
>
> Thank you for the suggestion. We are aware of more advanced MIAs such as LiRA, but these methods are not uniformly applicable across all three vision unlearning settings (classification, I2I, and T2I). To ensure a consistent and fair evaluation protocol across modalities, we adopt the shadow model–based MIA, which is an well-accepted approach that could generalize to all tasks. We agree that incorporating task-specific advanced MIAs is a valuable extension and plan to explore this in future work.
>
> **Weakness  5: Using existing attack method in T2I tasks weakens the contribution of this paper.**
> Thank you for the comment. UMA is designed as a general, task-agnostic modular post-unlearning attack, which can be instantiated differently depending on the target task. In the T2I case, when UMA is instantiated under the diffusion-model formulation, it naturally simplifies to a form equivalent to an existing T2I attack. We intentionally adopt this existing instantiation to demonstrate that our benchmark is compatible with and easily extensible to different adversarial probes, rather than tied to a single attack design. This compatibility is an important aspect of RUB’s contribution, ensuring that the benchmark can readily incorporate new T2I attacks developed by the community.
>
> **Question 1: For machine unlearning, why does retrain method remain almost unaffected under optimized malicious perturbations?**
>
> Thank you for the insightful question. The observation that the retrain method remains largely unaffected under optimized malicious perturbations is in fact expected. A retrained model contains no residue information from the forgotten data, and in semantic unlearning this “clean retrain” serves as the ideal target that approximate unlearning methods aim to match. Because there is no trace of the removed data for the attack to exploit, adversarial perturbations cannot meaningfully alter the model’s behavior.
>
> This differs from conventional adversarial attacks in classification, where the objective is to exploit decision-boundary ambiguities between classes. In contrast, our unlearning attack probes whether information traces of the forgotten data persist in the unlearned model. Since the retrained model truly lacks such traces, its outputs remain stable under attack. The gap between retrain and other unlearning methods therefore highlights the latter’s residual-information vulnerabilities, rather than indicating a limitation of the attack.
>
>
> **Question 2: Could the proposed benchmark be extended to include large language model unlearning scenarios?**
>
>
> In principle, yes. The benchmark concept can be extended to LLM unlearning. However, unlike vision models that operate in continuous latent spaces, LLMs are inherently discrete generative systems, which means that residual knowledge, attack implementations, and evaluation protocols take fundamentally different forms. As a result, applying RUB to LLMs would require modality-specific adaptations rather than a direct extension of the current framework.

---

### Official Review · Reviewer_9EoY · 2025-10-29

**Soundness:** 2
**Presentation:** 3
**Contribution:** 1
**Rating:** 2
**Confidence:** 4

**Summary:**

The paper proposes RUB, a benchmark and evaluation protocol for robust machine unlearning across three tasks: classification, image-to-image (I2I) inpainting, and text-to-image (T2I) diffusion. The underlying framework in RUB is the Unlearning Mapping Attack (UMA), a white-box post-unlearning attack instantiated per task (e.g., perturbing inputs or prompts) to probe for residual knowledge of the model about the forgotten samples/classes. Experiments span 15 unlearning methods; results suggest many pass standard verification but fail under adversarial probing.

**Strengths:**

1. The paper focuses on an important problem with machine unlearning methods which might be overlooked by many of the works that propose a new unlearning method.

2. By unifying the three tasks authors show that this problem exists for various tasks.

3. The paper touches on a timely problem as the introduced methods for machines have ramped up.

**Weaknesses:**

1. The paper is not up to date with the most recent unlearning methods and evaluation metrics. For example, MIA used for evaluation of unlearning in discriminative models were introduced in 2017, while several seminal and more effective methods have been published since then, look at [1,2] just to name a few. Some recent unlearning methods have adapted these methods to the setting of unlearning [3]. There have even been new MIA specifically designed for evaluating the effectiveness of unlearning [4,5]. Ignoring the whole advances in membership inference attacks in a newly available benchmark, not only does not help the community in advancing their unlearning methods, but will provide a wrong point of reference for the future works because they might build their evaluations based on this new benchmark despite the fact the the underlying metrics are outdated.

2. Even the unlearning methods are not updated and the most recent ones are published in early 2024. Considering that this work, if accepted, will be published in early 2026, that would be a very large gap and ignores the rapidly flourishing advances in unlearning methods and might drive the community back by providing a wrong point of reference for comparisons and evaluations. Just to name a few missing works, please see: [3,6,7,8,9,10,11]

3. The presented work lacks novelty, and most of it seems more like a concatenation of the following papers: [12,13]. More specifically, authors have used the exact attack methodology from [12] and [13] and even the experiments are very similar for T2I to [13]. It is expected to at least have a much more comprehensive set of experiments for a paper whose focus is merely on introducing a benchmark.

4. Given that the proposed threat model has been introduced by recent work in unlearning for both I2I and T2I tasks, one contribution of the presented work can be considered as unifying the threat model for the three tasks. However, I am not even sure if this unification is even necessary given that they come from different domains and each task requires its own specialized methodology and evaluation metrics and an author in any of these three domains would not need to look at the results for other tasks. For one of the domains I2I, which is newly introduced in [12] with only two of the methods applicable, is a benchmark even needed?

5. In ML literature when the reader sees propositions they expect some theoretical results with accompanying proofs. Here the propositions seem to be some general guidelines or hypotheses. And they are not novel as well and have been used in prior work to generate such attacks on unlearning models in T2I and I2I tasks.

6. In line 421 authors mention that in choosing the hyper-parameters the trade-off of TA with strength of unlearning has to be considered, and they do not mention how they decide about this trade-off? Maybe an average gap, similar to what has been reported in SalUn [14] would be a good combination of the metrics they use for evaluations.


[1] N. Carlini, S. Chien, M. Nasr, S. Song, A. Terzis and F. Tramèr, "Membership Inference Attacks From First Principles," 2022 IEEE Symposium on Security and Privacy (SP), San Francisco, CA, USA, 2022, pp. 1897-1914, doi: 10.1109/SP46214.2022.9833649.

[2] Zarifzadeh, S., Liu, P., & Shokri, R. (2024, July). Low-cost high-power membership inference attacks. In Proceedings of the 41st International Conference on Machine Learning (pp. 58244-58282).

[3] Ebrahimpour-Boroojeny, A., Sundaram, H., & Chandrasekaran, V. Not All Wrong is Bad: Using Adversarial Examples for Unlearning. In Forty-second International Conference on Machine Learning.

[4] Hayes, J., Shumailov, I., Triantafillou, E., Khalifa, A., & Papernot, N. (2025, April). Inexact unlearning needs more careful evaluations to avoid a false sense of privacy. In 2025 IEEE Conference on Secure and Trustworthy Machine Learning (SaTML) (pp. 497-519). IEEE.

[5] Cadet, X. F., Borovykh, A., Malekzadeh, M., Ahmadi-Abhari, S., & Haddadi, H. (2025, June). Deep Unlearn: Benchmarking Machine Unlearning for Image Classification. In 2025 IEEE 10th European Symposium on Security and Privacy (EuroS&P) (pp. 939-962). IEEE.

[6] Cywiński, B., & Deja, K. SAeUron: Interpretable Concept Unlearning in Diffusion Models with Sparse Autoencoders. In Forty-second International Conference on Machine Learning.

[7] Georgiev, K., Rinberg, R., Park, S. M., Garg, S., Ilyas, A., Madry, A., & Neel, S. (2025). Machine unlearning via simulated oracle matching. In The Thirteenth International Conference on Learning Representations.

[8] Zhang, B., Dong, Y., Wang, T., & Li, J. (2024, July). Towards Certified Unlearning for Deep Neural Networks. In International Conference on Machine Learning (pp. 58800-58818). PMLR.

[9] Wu, J., & Harandi, M. (2025). Munba: Machine unlearning via nash bargaining. In Proceedings of the IEEE/CVF International Conference on Computer Vision (pp. 4754-4765).

[10] Cha, S., Cho, S., Hwang, D., Lee, H., Moon, T., & Lee, M. (2024, March). Learning to unlearn: Instance-wise unlearning for pre-trained classifiers. In Proceedings of the AAAI conference on artificial intelligence (Vol. 38, No. 10, pp. 11186-11194).

[11] Bonato, J., Cotogni, M., & Sabetta, L. (2024, September). Is retain set all you need in machine unlearning? restoring performance of unlearned models with out-of-distribution images. In European Conference on Computer Vision (pp. 1-19). Cham: Springer Nature Switzerland.

[12] Li, G., Hsu, H., Chen, C. F., & Marculescu, R. Machine Unlearning for Image-to-Image Generative Models. In The Twelfth International Conference on Learning Representations.

[13] Zhang, Y., Jia, J., Chen, X., Chen, A., Zhang, Y., Liu, J., ... & Liu, S. (2024, September). To generate or not? safety-driven unlearned diffusion models are still easy to generate unsafe images... for now. In European Conference on Computer Vision (pp. 385-403). Cham: Springer Nature Switzerland.

[14] Fan, C., Liu, J., Zhang, Y., Wong, E., Wei, D., & Liu, S. (2023). Salun: Empowering machine unlearning via gradient-based weight saliency in both image classification and generation. arXiv preprint arXiv:2310.12508.

Minor weaknesses:

1. The work mentions unlearning accuracy has been used as the evaluation metric in some prior works. That is not the case in recent papers as they have updated their methodologies. The papers that only use unlearning accuracy are outdated and cannot be a point of comparison anymore.

2. Lower values of UA does not necessarily indicate better unlearning (line 255)

3. You also mention the setting of class unlearning in line 215 for discriminative models, but that is not a part of the presented benchmarks.

4. The subscript for function $f$ in $f_u(.,\theta^u)$ and $f_r(.,\theta^r)$ seems confusing as unnecessary because it initially conveys the idea that it shows the underlying data used for training, but then it is differently used for $f_u$ and it is just redundant subscript in current form.

5. The assumption in equation one about how the desired behavior of the unlearned model is misleading as it suggests that the expectation of the difference of output on the forget samples should become unboundedly larger. This is not a correct assumption as mentioned by prior work and might even lead to what is called overunlearning [1]

6. In proposition 2, it think you mean x+\delta_x not \delta_x? otherwise \delta_x seems to be overloaded by proposition 2 and 3 which is confusing for the reader.

7. “following prior arts”? In line 218


[1] Shi, W., Lee, J., Huang, Y., Malladi, S., Zhao, J., Holtzman, A., ... & Zhang, C. (2024). Muse: Machine unlearning six-way evaluation for language models. arXiv preprint arXiv:2407.06460.

**Questions:**

1. Are there any justification for choosing a MIA method published in 2017?

2. Will the authors be able to add the several missing recent methods in unlearning to their paper?

3. Details about the hyper-parameters of the unlearning methods and some details such as the number of iterations for PGD attack is missing. Would it be possible for the authors to include those details?

4. Are there any theoretical guarantees for the presented propositions that are missing in the current submission? Having the empirical evidence would be fine, but with the current wording it seems as if they are accompanied by theoretical proofs.

---

> ### Author Response · Authors · 2025-12-02
> **Response to Reviewer 9EoY**
>
> We thank the reviewer for the detailed feedback. Several of the comments, however, are based on **factual inaccuracies** or **misunderstandings** of our method. We address these issues and clarify the correct interpretations point by point below.
>
> **Comment 1: Why select shadow model-based MIA as the metric?**
>
> We are aware of more advanced MIAs such as LiRA, but these methods are not uniformly applicable across all three vision unlearning settings (classification, I2I, and T2I). As reflected in the referenced papers, advanced MIAs typically target one specific task, either discriminative scenarios or diffusion-based generation, and therefore cannot be applied consistently across modalities. Because our benchmark aims to provide a unified and fair evaluation protocol, we adopt the shadow model–based MIA, which is the only approach that generalizes to all tasks. We agree that incorporating modality-specific advanced MIAs is a valuable extension and plan to explore this in future work.
>
> **Comment 2: Is it possible to include more unlearning methods?**
>
> We appreciate the suggestion. Our benchmark is intended to be an ongoing and extensible resource rather than a closed set of experiments. We fully plan to incorporate additional unlearning methods as the field progresses. The codebase has been designed in a modular and plug-and-play manner, specifically to facilitate the easy integration of new algorithms into the benchmark.
>
> **Comment 3: The manuscript is a concatenation of previous unlearning study on [12-13].**
>
> We believe this assessment is based on a misunderstanding of the relationship between our work and [12–13]. It is true that both our benchmark and these prior studies include text-to-image unlearning tasks, and therefore there is some natural overlap in this specific setting. However, the goals, scope, and methodology are fundamentally different.
>
> References [12–13] focus narrowly on whether adversarial attacks can circumvent diffusion-based T2I unlearning. Our benchmark, in contrast, aims to evaluate whether residual information persists after unlearning across multiple vision modalities, which requires distinct evaluation protocols, metrics, and attack formulations. The datasets from [12–13] are included only to maintain consistency in the T2I case, not because our framework is derived from those works.
>
> More importantly, our benchmark is substantially broader, covering discriminative tasks, image-to-image reconstruction, and text-to-image synthesis within a unified and extensible framework. To the best of our knowledge, this is the first comprehensive benchmark for assessing robust unlearning in computer vision, and its contributions extend well beyond the scope of [12–13].
>
> **Comment 4: Is it necessary to introduce a unified benchmarking, since different tasks have different targets?**
>
> We strongly disagree with this comment. Although different tasks have different output targets, machine unlearning itself is a general learning objective, and the community increasingly seeks unlearning methods that are not restricted to a single task or modality. In fact, the gold-standard approach, model retraining, and recent methods such as SalUn already apply broadly across discriminative and generative settings, demonstrating that unified unlearning algorithms are both feasible and desirable.
>
> A unified benchmark is therefore important: it provides a consistent and comparable evaluation protocol across modalities, enabling the community to assess whether newly proposed unlearning methods truly generalize beyond narrow, task-specific scenarios. By offering such cross-task evaluation, our benchmark is designed to facilitate the development and adoption of more robust, task-agnostic unlearning approaches.
>
>
> **Comment 5: Theoretical guarantees for Proposition 2 and 3.**
>
> Thank you for the comment. Proposition 2 and 3 are characterizations of unlearning methods that fail to completely remove residue knowledge, rather than universal statements intended to hold for all possible unlearning algorithms. As our own experiments show, these propositions do not hold for the gold-standard unlearning targe, model retraining, because no residue information remains. For the approximate unlearning methods evaluated in our benchmark, however, we consistently observe that small perturbations are sufficient to resurface forgotten data, indicating that Proposition 2 and 3 accurately describe their behavior.

---

> > ### Author Response · Authors · 2025-12-02
> > **Response to Reviewer 9EoY (cont)**
> >
> > **Comment 6: Trade-off between unlearning utility and robustness.**
> >
> > We thank the reviewer for this insightful observation. We indeed observe that several approximate unlearning methods exhibit reduced task accuracy (TA) as their robustness against adversarial probing (RA) improves, reflecting how different algorithms balance utility and unlearning. However, this trade-off is not inherent to the unlearning problem itself: the retrain model achieves both high TA and strong robustness, demonstrating that utility and robustness are simultaneously attainable when unlearning is fully successful. Thus, the utility–robustness trade-off should be viewed as a limitation of current approximate unlearning methods, and resolving it is an important direction for future algorithmic development.

---

### Official Review · Reviewer_4Ln3 · 2025-10-31

**Soundness:** 3
**Presentation:** 3
**Contribution:** 2
**Rating:** 4
**Confidence:** 3

**Summary:**

The paper introduces a unified framework for assessing Machine Unlearning algorithms and specifically focusing on the robustness of these algorithms. The authors evaluate 15 state of the art unlearning algorithms across 3 types of tasks: classification, image-to-image reconstruction and text-to-image generation. The authors also introduce a unified attack mechanism called Unlearning Mapping Attack (UMA) which employs worst-case white-box adversary, granting full knowledge of the models and forget sets to modify the inputs to the models only. The key findings are that all the state of the art unlearning methods are fragile to adversarial perturbations.

**Strengths:**

- The paper identifies an important failure case of the current state of the art machine unlearning algorithms.
- Extensive experiments across a variety of domains (classification, I2I, T2I) and unlearning algorithms (15 algorithms).
- Easy to read and follow.

**Weaknesses:**

- Limited technical novelty as UMA seems like essentially the same as standard gradient based adversarial attack.
- There are already adversarial attacks and robust machine unlearning methods for LLMs.

**Questions:**

- Why did the authors not include LLMs?
- For the T2I attacks, can the authors provide some examples of what the adversarial text input looks like?

---

> ### Author Response · Authors · 2025-11-30
> **Response to Reviewer 4Ln3**
>
> We would like to thank the reviewer for the comments. Below is our point-to-point response.
>
> **Weakness 1: Limited technical innovation of the adversarial attack.**
>
> The novelty of our work does not lie in proposing a new attack primitive. Instead, it lies in establishing the first unified benchmark, evaluation protocol, and codebase for assessing post-unlearning adversarial recovery across classification, I2I, and T2I tasks. UMA is a general-purpose verification framework, not merely a gradient attack: it provides a task-agnostic interface that enables consistent, cross-domain evaluation of residual knowledge, a capabilities that standard gradient attacks do not offer. The benchmark is explicitly designed to support plug-in adversarial probes, and we demonstrate this by instantiating multiple attack variants beyond UMA. Thus, the primary contribution is the systematic, cross-modal evaluation framework that exposes fundamental limitations of current unlearning methods, rather than the specific attack formulation itself.
>
> **Weakness 2 and Question 1: Why not include LLM in the benchmark.**
>
> We focus on vision tasks because the inherent nature of vision and text is fundamentally different: vision models operate in continuous latent spaces, whereas LLMs rely on discrete token representations. These differences produce distinct forms of residual knowledge and require different unlearning mechanisms and evaluation protocols, making a unified benchmark impractical at this stage. While LLMs already have emerging robustness surveys and evaluation efforts, we note that no such fair, systematic comparison framework exists for vision unlearning, and our study fills this gap.
>
> **Question 2: Examples of adversarial text for T2I tasks**
>
> T2I adversarial prompts are generated by injecting disruptive token sequences before the original text prompt to perturb the model’s conditioning. For example, for the original prompt “old stone church in countryside”, two adversarial examples generated in our experiments are “tman vere shetland old stone church in countryside and “imransubscription mahon old stone church in countryside” These prepended tokens may not be semantically meaningful but effectively disturb the text-conditioning pathway, enabling the adversarial probe to expose residual knowledge.

---

### Meta-Review · Area_Chair_G3tX · 2026-01-05

**Summary:**

In the submission, a model unlearning benchmark with a residual information detection method was proposed. Experiments are conducted on computer vision related tasks. While all reviewers acknowledged that the problem is important, all reviewers raised concerns that the scope of the submission could be limited, since no text-related tasks and LLMs are evaluated. Reviewers also pointed out the proposed benchmark could be limited, which is a combination of existing works, the threat model of the benchmark, as well as the evaluation metrics of the benchmark.

**Reviewer Concerns:**

The authors provided rebuttal but no reviewers respond. The AC also read through the rebuttal and manuscript, and found the response is still lacking, especially regarding the scope of the benchmark, where the authors claimed the benchmark to be unified to only three computer vision tasks are evaluated.

**Reviewer Scores:**

Reviewers are highly unlikely to increase the scores.

---

### Decision · Program_Chairs · 2026-01-26

Reject